# A Review on Injection Molding: Conformal Cooling Channels, Modelling, Surrogate Models and Multi-Objective Optimization

**DOI:** 10.3390/polym17070919

**Published:** 2025-03-28

**Authors:** António Gaspar-Cunha, João Melo, Tomás Marques, António Pontes

**Affiliations:** Department of Polymer Engineering, University of Minho, 4710-057 Braga, Portugal; pg50466@alunos.uminho.pt (J.M.); pg50789@alunos.uminho.pt (T.M.); pontes@dep.uminho.pt (A.P.)

**Keywords:** injection molding, design of molds, conformal cooling channels, surrogate models, multi-objective optimization

## Abstract

Plastic injection molding is a fundamental manufacturing process used in various industries, accounting for approximately 30% of the global plastic product market. A significant challenge of this process lies in the need to employ sophisticated computational techniques to optimize the various phases. This review examines the optimization methodologies in injection molding, with a focus on integrating advanced modeling, surrogate models, and multi-objective optimization techniques to enhance efficiency, quality, and sustainability. Key phases such as plasticizing, filling, packing, cooling, and ejection are analyzed, each presenting unique optimization challenges. The review emphasizes the importance of cooling, which accounts for 50–80% of the cycle time, and examines innovative strategies, such as conformal cooling channels (CCCs), to enhance uniformity and minimize defects. Various computational tools, including Moldex3D and Autodesk Moldflow, are discussed due to their role in process simulation and optimization. Additionally, optimization algorithms such as evolutionary algorithms, simulated annealing, and multi-objective optimization methods are explored. The integration of surrogate models, such as Kriging, response surface methodology, and artificial neural networks, has shown promise in addressing computational cost challenges. Future directions emphasize the need for adaptive machine learning and artificial intelligence techniques to optimize molds in real time, offering more innovative and sustainable manufacturing solutions. This review is a comprehensive guide for researchers and practitioners, bridging theoretical advancements with practical implementation in injection molding optimization.

## 1. Introduction

Plastic injection molding is one of the most critical manufacturing processes, responsible for approximately 30% of all plastic products produced globally [1,2,3]. It is widely used across various industries to manufacture complex, high-precision components, including those for the automotive, electronics, medical, and consumer goods sectors. The process comprises multiple stages, such as plasticizing, injection, cooling, and ejection, each presenting unique challenges and optimization opportunities.

This review paper explores the optimization strategies used in plastic injection molding, focusing on integrating advanced algorithms, numerical modeling, surrogate models, and multi-objective optimization algorithms to enhance process efficiency, product quality, and sustainability [1,3]. Optimizing injection molding is crucial for reducing manufacturing costs, enhancing product quality, and minimizing cycle times. The scope of optimization spans different stages of the process, involving various decision variables and objectives such as reducing warpage, minimizing cooling time, and improving the mechanical properties of the final product [4]. Given the complexity and interdependence of these variables, sophisticated algorithms capable of handling multi-objective optimization problems with conflicting objectives are required [5,6].

Recent advancements in computational modeling tools, including Moldex3D, Autodesk Moldflow, and other numerical modeling software, have transformed how engineers design and optimize injection molds, given the design improvements enabled by tools like Moldex3D or Autodesk Moldflow and their impact on cycle times or defect reduction [7]. These tools enable the numerical modeling of the molding process, providing insights into the flow, pressure, and temperature distributions within the mold due to their computational efficiency. Such simulations are instrumental in optimizing mold design, particularly in implementing strategically conformal cooling channels (CCCs). CCCs follows the contour of the mold cavity, resulting in more uniform cooling and significantly improved cycle times and product quality [8,9].

Optimization procedures applied in injection molding are diverse, ranging from traditional methods like empirical approaches and gradient-based algorithms to more sophisticated ones such as evolutionary algorithms (EAs), simulated annealing (SA), particle swarm optimization (PSO), and multi-objective optimization (MOP) [8]. Each algorithm has unique strengths and limitations. For example, while EA is well-suited for global optimization, it often requires longer computational times and is highly sensitive to parameter settings. Additionally, these algorithms require evaluating many solutions during the search procedure, implying the need for surrogate models to replace the computationally expensive calculations.

Sustainability in injection molding optimization can be enhanced through several strategies, such as reducing cycle time to lower energy usage per part, minimizing waste by decreasing the occurrence of defects, mainly through improved cooling efficiency, optimizing part design to use less raw material without compromising functionality, and adopting process parameters that reduce overall energy consumption. The use of multi-objective optimization algorithms can enable the simultaneous improvement of these objectives.

This review describes these optimization algorithms by analyzing their applicability, strengths, and weaknesses in the context of injection molding. Previous work by Gaspar-Cunha et al. [10] provides a framework for employing evolutionary algorithms to address multi-objective optimization in plastic manufacturing processes, including injection molding. Furthermore, we will discuss the role of surrogate models, such as response surface methodology (RSM), Kriging, and artificial neural networks (ANN), which are often used to approximate complex simulations and reduce computational costs during optimization [11].

As the industry advances, further research is necessary to enhance the integration of machine learning and artificial intelligence into mold design optimization. Future directions may involve exploring adaptive algorithms that leverage data generated from numerical modeling during the optimization process, allowing for real-time process adjustments and more innovative mold design [12,13]. This review also analyzes the latest reviews and studies on CCCs and injection molding optimization methodologies. Despite significant progress, gaps remain in addressing the limitations of existing optimization methodologies and in selecting the optimized objectives.

Related reviews have explored various facets of CCCs and injection molding optimization, focusing on specific aspects such as cycle time reduction, cooling efficiency, or additive manufacturing techniques.

Shayfull et al. [14] highlighted the potential of CCCs in rapid heat cycle molding (RHCM) to reduce cycle times and enhance part quality, focusing on additive manufacturing and simulation tools for optimization. However, they primarily addressed cycle time reduction, overlooking broader objectives such as temperature profiles that impact defect formation, and thus limited the comprehensive optimization of CCCs.

Fernandes et al. [15] discussed optimization methodologies such as design of experiments (DOE), artificial neural networks (ANNs), and genetic algorithms (GAs) for injection molding. While evaluating strengths and limitations, they focused on conventional objectives like cycle time and product quality, lacking exploration of temperature distribution and defect prevention for a holistic optimization framework.

Kanbur et al. [8] explored additive manufacturing for CCC geometries using Computer-Aided Engineering (CAE) tools to optimize cooling performance. Most reviewed works focused on cooling efficiency and cycle time, neglecting critical factors such as shrinkage and warpage that affect CCC performance.

Wei et al. [16] reviewed CCC development and optimization using additive manufacturing methods like Selective Laser Melting (SLM) and Direct Metal Laser Sintering (DMLS). They discussed parametric optimization techniques to improve cooling efficiency but gave limited attention to mechanical integrity and defect minimization, which are essential for consistent part quality.

Feng et al. [17] reviewed CCC design and hybrid additive–subtractive manufacturing methods to improve dimensional accuracy. However, they lacked emphasis on practical applications or real-world case studies, limiting insights into the feasibility of their proposed methodologies.

Silva et al. [18] examined multi-objective optimization strategies for CCCs using advanced additive manufacturing and simulation tools to balance cooling efficiency and mechanical integrity. Despite these efforts, the reviewed methodologies often focused on a limited set of objectives, failing to address the complexities of balancing conflicting goals.

The current literature on CCCs and IM optimization primarily targets a narrow set of objectives like cycle time, shrinkage, warpage, and energy consumption. Also, there is a lack of a framework for comparing optimization methods and a new point of view that addresses both theoretical advancements and practical implementation gaps, proposing comprehensive optimization approaches:
Broader Optimization Objectives: Future work should go beyond conventional objectives, including new considerations like temperature profiles linked to defect formation for comprehensive optimization.Practical Applications: Practical case studies are needed to bridge the gap between theory and practice.Multi-Objective Optimization: New approaches should balance conflicting objectives, such as cooling efficiency, mechanical integrity, and defect prevention.Data Mining: Using data mining tools is fundamental to understanding the link between the data involved (decision variables and objectives) and building surrogate models.Simplified Guidelines: Developing accessible guidelines for adopting advanced optimization techniques is crucial for broader adoption by practitioners.

An evident trend in the body of research on injection molding optimization is the compartmentalized treatment of distinct objectives, such as the design of conformal cooling channels (CCCs), as independent challenges. Despite the availability of advanced methodologies, including multi-objective evolutionary algorithms (MOEAs) and surrogate modeling techniques like artificial neural networks (ANNs), which have demonstrated considerable success in optimizing processing parameters, these approaches are seldom leveraged when addressing new or parallel objectives. Instead, many studies regress to elementary strategies, frequently relying on single-objective formulations and classical design of experiments (DOE), thereby reinitiating the optimization framework without building upon prior methodological progress. This review aims to highlight this fragmentation and advocates for a more coherent and cumulative application of advanced optimization paradigms across the various facets of the injection molding process.

Therefore, this review aims to serve as a comprehensive guide for researchers and practitioners alike, aiding in the selection of the most appropriate optimization methodologies for designing molds, with a special emphasis on molds utilizing conformal cooling channels. By examining current methodologies and discussing future directions, this paper contributes to the ongoing advancement of plastic injection molding technology.

This article is organized as follows: Section 2 explores the concepts and optimization methods related to the injection molding process as presented in the existing literature. Section 3 presents a literature review concerning the global process and the CCCs. Section 4 highlights the most important conclusions and provides suggestions for further work.

## 2. Optimization of the Injection Molding Process

### 2.1. Injection Molding Cycle

The injection molding process comprises several critical phases: plasticating, filling, packing, cooling, and ejection (Figure 1A). Each phase significantly influences the final quality of the molded part [1,4,19,20].

In the plasticating phase (i), polymer granules are heated and melted within the barrel of the injection molding machine. The temperature and shear conditions during plasticating are crucial as they affect the polymer’s viscosity and flow characteristics. In the filling phase (ii), the molten polymer is injected into the mold cavity through the sprue, runners, and gates. The design of the feed system, including the geometry of runners and gates, plays a vital role in achieving uniform flow and minimizing pressure drops. The packing phase (iii) aims to compensate for material shrinkage as the polymer cools. The duration and pressure of the packing phase are critical parameters that influence the final part’s density and mechanical properties. The cooling phase (iv) allows the molten polymer to solidify within the mold. The design and efficiency of the cooling system, including the placement and geometry of the cooling channels, are crucial for achieving uniform cooling. Non-uniform cooling can lead to residual stresses, warpage, and dimensional inaccuracies. Advanced cooling techniques, such as conformal cooling channels that follow the mold’s contours, have been developed to enhance cooling efficiency and part quality. Finally, in the ejection phase (v), the part is ejected from the mold when it is sufficiently cooled and solidified. Proper draft angles and surface finishes on the mold surfaces facilitate smooth ejection.

Simultaneously, by systematically analyzing and optimizing each phase of the injection molding cycle, manufacturers can enhance part quality, reduce cycle times, and improve overall process efficiency. The duration of each stage is dependent on the material properties and the specific design characteristics of the part. Consequently, optimizing these stages is imperative for enhancing production efficiency [1,4]. Among these stages, cooling is often the most time-intensive, comprising between 50% and 80% of the overall cycle time, as shown in Figure 1B [18]. Therefore, achieving efficient heat transfer is crucial for minimizing cooling duration while maintaining part quality and ensuring dimensional accuracy.

### 2.2. Optimization Characteristics

Optimizing the injection molding process involves fine-tuning parameters across various phases to minimize defects such as shrinkage, warpage, and residual stresses. Critical factors include:
Material Temperature: The initial temperature during the plasticating phase affects polymer viscosity and flow behavior, influencing the subsequent process phases.Process Parameters: The injection speed, pressure, packing pressure, and cooling time must be carefully optimized for specific materials and part geometries to ensure consistent quality.Feed System and Cooling Channel Design: The geometry of runners, gates, and cooling channels governs flow patterns and cooling rates, directly impacting the final part’s microstructure, mechanical properties, and dimensional stability.Mold Temperature: Uniform and properly controlled mold temperatures are crucial for consistent cooling, solidification, and dimensional accuracy.

Understanding the interplay between polymer flow dynamics and thermal profiles during the plasticating, filling, packing, cooling, and ejection phases is crucial. These factors dictate the development of the part’s microstructure, ultimately determining its mechanical properties and dimensional precision. Poor control of these parameters can lead to defects such as shrinkage, warpage, and residual stresses, compromising product quality [4].

Figure 2 illustrates the interconnections between decision variables, such as process parameters and system geometry, the various phases of the injection molding cycle, and the objectives to be optimized. Mold temperature, for instance, plays a central role across most phases, conditioning the process behavior and influencing the final parts’ quality [20].

Selecting appropriate optimization objectives presents a significant challenge due to the complexity of the thermomechanical phenomena involved simultaneously. Flow and temperature dynamics influence the development of microstructure, which in turn affects potential defects such as shrinkage, warpage, and residual stresses. However, the interdependency between temperature dynamics and these defects remains inadequately understood. An open question is whether optimization objectives should directly measure temperature dynamics or focus instead on defective outcomes.

Each phase of the injection molding cycle impacts different aspects of the final part’s quality, as illustrated in Figure 2: The plasticating phase determines the initial melt temperature, influencing viscosity and flow behavior; the filling phase defines the flow dynamics, affecting molecular orientation and microstructure; the packing phase influences part density and residual stresses; the cooling phase establishes temperature dynamics and dimensional stability; and the ejection phase affects part distortion and surface finish.

Various performance measures can serve as potential optimization objectives in each phase. These measures, however, often conflict, making the optimization process inherently multi-objective and complex. Determining whether it is better to prioritize thermal dynamics or defect minimization in objective selection remains a key research challenge [21].

### 2.3. Design Variables

The design variables in injection molding play a crucial role in determining the efficiency, quality, and overall success of the manufacturing process. These variables are broadly categorized into four main domains: operating conditions, feed system, cooling channels, and ejection system. Effective optimization of these elements is critical for achieving superior product quality, enhanced throughput, and cost-effectiveness [22].

Operating conditions (Figure 1) encompass several key parameters that directly influence the molding process and product outcomes [19,23,24].

Melt Temperature (*T_melt_*): The temperature of the molten polymer affects viscosity, flow behavior, and material properties. Precise control ensures proper mold filling and reduces degradation risks.Mold Temperature (*T_mold_*): The mold temperature has a critical influence on cooling rates, cycle times, and product quality. Elevated mold temperatures enhance the surface finish and reduce residual stresses but may extend cycle durations. Conversely, lower temperatures expedite cooling but can lead to defects such as warpage and shrinkage.Ejection Temperature (*T_eje_*): The temperature at which the part is ejected from the mold impacts dimensional stability and surface integrity.Coolant Temperature (*T_coolant_*) and Air Temperature (*T_air_*): These temperatures directly affect the cooling system’s efficiency, and the coolant temperature must be optimized for consistent thermal management.Injection Time (*t_inj_*): This parameter determines the duration for which material is injected into the mold. Proper timing ensures uniform filling and mitigates air entrapment.Packing Time (*t_pack_*): Essential for compensating material shrinkage during cooling, optimized packing time prevents voids and sink marks.Cooling Time (*t_cooling_*): This significantly impacts cycle time and productivity. Reducing cooling time while maintaining part integrity is key to efficient operation.Injection Speed (*V_inj_*): The velocity of polymer flow during injection must be optimized to ensure consistent filling and avoid air traps or material shear.Injection Pressure (*P_inj_*) and Packing Pressure (*P_pack_*): These pressures ensure mold filling and compensate for shrinkage. Excessive pressure, however, can lead to defects such as warpage and material stress.

The geometry of cooling channels is crucial for effective thermal management during the injection molding process. Well-designed cooling systems enhance heat dissipation, reduce cycle times, and ensure dimensional stability. Figure 3 illustrates an example where the goal is to design two alternative situations: one with two circular rings and the other with three circular rings. In these cases, key geometric parameters include:
Diameter (D1, D2, and D3): The diameter of the cooling channels determines the flow rate and cooling efficiency. Larger diameters enhance cooling but may reduce mold strength.Pitch Distance (d1, d2, and d3): The spacing between adjacent channels, also known as the pitch distance, affects the cooling uniformity. A pitch that is too large may lead to uneven cooling and warpage, while a pitch that is too small can weaken the mold structure.Distance Between Channel Centers and Part Surface (d’1, d’2, and d’3): The distance from the center of the cooling channel to the part surface must be optimized for effective thermal transfer without compromising the part’s structural integrity.Length: The overall length of the cooling channels influences the temperature gradient and pressure drop across the system.

The placement of cooling channels should ensure uniform thermal distribution, minimal pressure drop, and enhanced mold longevity. Advanced cooling techniques, such as conformal cooling, have been increasingly employed to achieve superior thermal performance by tailoring channel geometry to the part’s complex shape [25].

The mold geometry, encompassing gate locations, runner dimensions, and cooling channels, is crucial in achieving uniform flow, reducing pressure drops, and improving cooling efficiency [1]. The ejection system, including the placement and design of ejector pins, ensures the smooth removal of parts without damage or deformation.

### 2.4. Optimization Objectives

In general, the objectives to be optimized are related to (i) reducing common defects such as shrinkage, warpage, sink marks, voids, and incomplete filling; (ii) reducing cycle time, essential for improving production efficiency and reducing costs; (iii) enhancing mechanical properties such as strength, flexibility, and impact resistance based on the intended use of the product; (iv) improve surface finish, ensuring the final product meets esthetic requirements without further processing; (v) optimize material use, reducing waste and improving the overall sustainability of the manufacturing process.

Due to its importance, it is relevant to stress here that injection molding is susceptible to various defects that can potentially affect product quality, which can be difficult to overcome. For this reason, these defects are commonly studied in the literature and industrial practice. For example, most of these defects derive from the cooling phase [1,4,26]. Due to uneven cooling rates across the part, the warpage can result in differential shrinkage, causing the part to deform or warp. Variations in wall thickness, geometric features, or insufficient cooling time contribute to warpage. Sink marks, resulting from differential cooling rates, can lead to uneven shrinkage, causing depressions or dimples on the part’s surface. These sink marks often occur in thick sections where cooling is slower. Internal stress and cracking due to rapid or uneven cooling can result in internal stresses within the part. These stresses may exceed the material’s strength, leading to cracking or fracture. Warpage can also induce internal stresses. Short shots occur when the mold cavity is incompletely filled due to premature solidification, resulting in parts that are either incomplete or undersized. Shrinkage due to non-uniform cooling can lead to variations in part dimensions across the mold, resulting in dimensional inaccuracies. This defect is particularly significant in large or complex parts.

However, two aspects of optimizing the process are particularly challenging to address. Firstly, most of these defects are difficult to quantify, making it impossible for any numerical modeling software to calculate them. The alternative is to determine it experimentally and find a way of considering it for optimization, using, for example, artificial neural networks or another regression technique. Secondly, in most cases, it is impossible to determine a causal connection between the defect and the decision variables. This difficulty is related to the previous one, since if this causal effect is not known, it is impossible to calculate the extension of the defect [27]. Simultaneously, the different software can calculate the values of essential defects, such as shrinkage, warpage, short shot, and internal stresses [28].

Figure 4 illustrates an example where the objective is to minimize the thickness distribution difference between the inner and outer lines of a plastic part (Figure 4A), thereby achieving a uniform thickness throughout the entire wall. Figure 4B presents two potential temperature distribution profiles alongside the target value. When the goal is to align the profiles as closely as possible with the target value, and both profiles share the same average value, Profile 2 appears more favorable. This demonstrates that assessing only the average value of all points on the surface is inadequate; evaluating the deviations of the most extreme points is equally important. Furthermore, it is worth emphasizing that most defects stem from non-uniform cooling. As such, incorporating temperature profiles in defect prevention is highly significant. This introduces an additional challenge to the optimization process, as incorporating more objectives increases complexity and necessitates using tools like principal component analysis (PCA) to identify and select the most relevant objectives.

### 2.5. Optimization Methodologies

Optimization methods in injection molding are diverse and can be categorized based on their objectives and approaches. These include single-objective, multi-objective, and hybrid methods, as well as data-driven and AI-driven techniques. Below is an overview of the most prominent methods, along with their applications and characteristics, explained in a more accessible manner.

Traditional Methods:
Empirical Methods: These rely on trial and error, using prior experience to refine settings for injection molding. They are straightforward but can be time-consuming.Design of Experiments (DOE): DOE systematically examines the influence of input variables on outputs, optimizing conditions by statistically analyzing multiple factors simultaneously [29].Taguchi Method: This approach optimizes parameter settings to minimize variability and improve quality, emphasizing robust design [30].Taguchi with Gray Relational Analysis (GRA): This method addresses problems with multiple performance metrics by combining Taguchi’s robustness with GRA, ensuring comprehensive optimization [31].

Iterative Optimization Techniques [32]:Simplex Method: Designed for linear problems, this mathematical technique efficiently navigates through feasible solutions to find the optimal one.Complex Method: Suitable for non-linear and multimodal functions, this approach identifies global or near-global optima in complex scenarios.Gradient Methods: Gradient-based techniques, such as gradient descent, utilize derivative information to optimize smooth, differentiable problems efficiently.Direct Search Methods: These are ideal for non-smooth or discontinuous problems, as they do not rely on derivative information.

Advanced Computational Methods:Sequential Approximate Optimization (SAO): SAO creates surrogate models to approximate objectives, enabling faster optimization of complex problems [33].Sequential Quadratic Programming (SQP): This method solves a series of quadratic subproblems, excelling in optimization scenarios with non-linear constraints [34].

Stochastic and Evolutionary Approaches:Evolutionary Algorithms (EA): Inspired by natural selection, EAs can solve complex, multi-variable, and global optimization problems [35].Particle Swarm Optimization (PSO) and Multi-Objective PSO (MOPSO): These algorithms simulate social behavior to solve non-linear, multi-dimensional problems efficiently [36].Simulated Annealing (SA) is a probabilistic method for exploring the solution space to approximate the global optimum in complex problems [37].

Multi-Objective Optimization Techniques:Multi-Objective Evolutionary Algorithms (MOEAs): Adaptations of EA to address multi-objective optimization problems based on non-dominance. For example, the NSGA-II (Non-Dominated Sorting Genetic Algorithm II) is a popular MOEA that handles trade-offs between conflicting objectives [38].Multi-Objective Firefly Algorithm (MOFA): Inspired by firefly behavior, MOFA effectively solves multi-objective problems with strong convergence properties [39].Multi-Objective Bayesian Optimization (MBO): MBO balances exploration and exploitation using Bayesian inference, reducing computational effort [40].

Topology Optimization

Topology Optimization (TO): A computational method used to design structures or materials by optimizing their layout within a given design space to achieve the best performance while satisfying constraints [41].

Data-Driven, AI-Driven, and Fuzzy Logic Approaches

Data and AI-Driven Optimization: This methodology uses data, such as ANN, to drive the optimization method [42].Fuzzy Optimization: This method incorporates fuzzy logic to address uncertainties and imprecise inputs, ensuring robust outcomes [43].

These methods highlight the breadth of techniques available for optimizing injection molding processes, from empirical approaches to sophisticated AI-driven models tailored to various complexities and objectives. They vary in complexity, each offering distinct advantages. Traditional methods, such as empirical approaches and Taguchi methods, rely on past experiences but can be time-consuming. Design of Experiments (DOE) enhances efficiency through systematic analysis of factors. Iterative techniques, such as the Simplex and gradient-based approaches, optimize structured problems but struggle with complexity. Direct search methods are particularly effective for non-smooth problems. Advanced computational techniques, such as Sequential Approximate Optimization (SAO) and Sequential Quadratic Programming (SQP), improve convergence but require precise models. Stochastic and evolutionary algorithms, including Evolutionary Algorithms (EAs) and Particle Swarm Optimization (PSO), excel in global optimization. Multi-objective approaches balance conflicting objectives, such as Multi-Objective Evolutionary Algorithms (MOEAs) and Bayesian Optimization (BO). Data-driven and AI-based methods, such as artificial neural networks (ANNs) and fuzzy logic, enhance adaptability and precision. Integrating AI with traditional optimization presents a promising approach to enhancing the efficiency of injection molding processes.

For example, Figure 5 illustrates the flowchart of a multi-objective evolutionary algorithm (MOEA) grounded in the concept of non-dominance, which plays a crucial role in guiding the population toward the Pareto frontier over successive generations [5,6].

Consider a problem with two objectives to be maximized (Figure 5A). Solutions located in the top-right region outperform the central solution in both objectives, rendering the central solution dominated by all solutions in this region. Conversely, solutions in the bottom-left region are worse in both objectives and are, therefore, dominated by the central solution. The set of solutions is not dominated by any other forms of the Pareto front, representing the optimal trade-offs among the objectives.

Non-dominance ensures solutions are ranked by their ability to balance trade-offs across objectives. Solutions are grouped into non-dominated fronts, with the first front prioritized to maintain diversity and drive progress toward the Pareto frontier, followed by subsequent fronts, which consist of solutions dominated by higher-ranking ones.

Non-dominance is crucial in multi-objective optimization, as it balances trade-offs without relying on arbitrary weights. By recombining top solutions, the algorithm generates a diverse and high-quality approximation of the Pareto front [5,6]. This concept can be applied with an EA (Figure 5B) to evolve a population of solutions toward the Pareto front across successive generations, thereby balancing the exploration and exploitation of the search space. In this case, the assignment of the fitness step considers the level of non-dominance of the solutions to improve their quality, thereby enabling them to be better selected for reproduction in the selection step.

### 2.6. Numerical Modeling and Surrogates

The optimization of the injection molding process involves a combination of experimental techniques, numerical simulations, and advanced computational models. Each approach makes a unique contribution to enhancing efficiency, quality, and cost-effectiveness. Below, we explore these methods in detail, highlighting their applications and advantages.

Experimental approaches are fundamental in observing and measuring real-time parameters during injection molding. These techniques are indispensable for validating numerical models and providing crucial temperature, pressure, and cooling data. By ensuring alignment with real-world conditions, experimental methods form the foundation for optimizing process parameters [44].

Simulation tools are crucial in predicting outcomes and refining designs before manufacturing. Three leading software solutions—Moldflow, Moldex3D, and ANSYS—offer distinct capabilities tailored to various aspects of injection molding analysis. This comparative analysis examines their strengths, applications, and key differentiators.

**Moldflow** [45]

Moldflow, developed by Autodesk, is widely used for simulating various aspects of the injection molding process, including (i) Filling, packing, cooling, and warpage analysis; (ii) prediction of defects such as air traps, sink marks, weld lines, and voids; (iii) optimization of gate and runner design to enhance material flow efficiency; (iv)support for a broad range of materials, providing accurate material behavior predictions; and (v) integration with CAD tools, enabling seamless design iterations. Moldflow is particularly useful for engineers seeking to minimize waste, reduce defects, and enhance part manufacturability.

**Moldex3D** [46]

Moldex3D is recognized for its ability to handle complex part geometries and accurately simulate melt behavior. Key features include the following: (i) advanced 3D numerical simulation for highly accurate predictions of material behavior; (ii) detailed insights into melt front behavior, cooling efficiency, and potential defects; (iii) improved cycle time estimation, helping manufacturers optimize production schedules; (iv) comprehensive material database, supporting accurate analysis of various thermoplastics; and (v) integration with multi-physics analysis, enabling the study of interactions between thermal and mechanical properties. Moldex3D is ideal for applications where precision and detailed analysis of complex geometries are essential.

**ANSYS** [47]

ANSYS offers a comprehensive range of engineering simulations, with a strong emphasis on thermal and structural analysis. Key features include the following: (i) structural and mechanical stress analysis, allowing for in-depth study of deformations during molding; (ii) optimization of material properties to enhance durability and performance; (iii) coupled thermal and flow analysis, ensuring realistic heat transfer modeling; (iv) finite element analysis (FEA) capabilities, aiding in mechanical performance evaluation; and (v) compatibility with other ANSYS modules, enabling multi-disciplinary simulations. ANSYS is best suited for engineers who require a comprehensive analysis of structural and thermal stresses, extending beyond the injection molding process.

Table 1 stresses the most important characteristics of each one of these programs.

**Table 1 polymers-17-00919-t001:** Comparative analysis of the injection molding software.

Feature/Software	Moldflow	Moldex3D	ANSYS
Primary Focus	Injection molding process simulation	Detailed melt flow and cooling analysis	Structural and thermal analysis
Filling, Packing, and Cooling	Excellent	Superior	Limited
Warpage and Defect Prediction	High accuracy	Very detailed	Moderate
Complex Geometry Handling	Moderate	Excellent	Limited
Structural and Stress Analysis	Basic	Moderate	Excellent
Material Behavior Simulation	Extensive database	Very precise	Advanced mechanical properties
Integration with CAD/CAE	Strong	Strong	Extensive (especially with the ANSYS ecosystem)
Best Use Case	Process optimization and defect minimization	Complex geometries and cycle time improvement	Structural and thermal performance evaluation

Choosing the right injection molding software depends on the specific needs of the project:**Moldflow** is best suited for general injection molding simulations, particularly for optimizing process parameters and minimizing defects.**Moldex3D** is ideal for detailed melt flow analysis and handling complex part geometries with high precision.**ANSYS** is more suitable for engineers who require in-depth structural and thermal analysis beyond the molding process itself.

Each software has its strengths, and in some cases, a combination of these tools may provide the most comprehensive analysis for optimizing injection molding processes.

In some cases, it can be beneficial to utilize custom software solutions, which are characterized by the development of tailored in-house models designed to address the specific needs of unique molds or parts. These bespoke tools combine experimental data with simulations for higher accuracy and reliability [48,49].

Integrating modeling with optimization routines is a common approach to effectively optimizing the injection molding process. However, the computational cost of evaluating numerous solutions can be prohibitive. To address this challenge, surrogate models or metamodels are employed, simplifying complex simulations while maintaining accuracy. Some important techniques include [50]:
Polynomial Regression (PR): Models variable relationships using polynomial equations [51].Response Surface Methodology (RSM): Utilizes statistical approaches to model interactions between variables [52].Kriging: A spatial interpolation method that creates surrogate models for high-dimensional data [53].Support Vector Machines (SVM) + Linear Regression: Combines SVM with linear regression to enhance prediction accuracy [54].Artificial Neural Networks (ANN): Leverage input data to predict outcomes and are often paired with Genetic Algorithms (GA) for enhanced optimization [55].Bayesian Methods: Incorporates probability distributions to quantify model uncertainty and enhance predictions [56].Radial Basis Function (RBF): Uses neural network methods to approximate multivariable functions [57].Quadratic Response Surface (QRS): Applies quadratic polynomial models for response surface analysis [58].Physics-Informed Neural Networks (PINN): Integrates physical laws into neural networks for more accurate modeling [59].Gaussian Process Regression (GPR) + ANN: Combines GPR and ANN for improved prediction reliability [60].Proper Orthogonal Decomposition (POD) + Polynomial Chaos Expansion (PCE): POD reduces dimensionality, while PCE approximates uncertainty propagation [61].

The abovementioned techniques encompass a range of statistical, machine learning, and physics-informed methods for modeling complex systems. Polynomial regression (PR) and quadratic response surface (QRS) offer simple yet limited approximations, struggling with high-dimensional data. Response surface methodology (RSM) extends polynomial models by capturing variable interactions but remains constrained by underlying assumptions. Kriging and Gaussian Process Regression (GPR) offer flexible, non-parametric interpolation methods well-suited for high-dimensional spaces, albeit at a computational expense. Machine learning approaches, such as support vector machines (SVMs), artificial neural networks (ANNs), and radial basis functions (RBFs), excel at capturing non-linear relationships; however, they require substantial training data and fine-tuning. Physics-informed neural networks (PINNs) enhance predictive accuracy by incorporating physical constraints, making them well-suited for scientific modeling. Bayesian methods enhance uncertainty quantification, while proper orthogonal decomposition (POD) and polynomial chaos expansion (PCE) facilitate dimensionality reduction and uncertainty analysis, proving particularly useful in complex simulations. The effectiveness of each technique depends on the problem’s structure, data availability, and computational resources.

By combining these experimental, simulation, and surrogate modeling techniques, engineers can achieve a more efficient, reliable, and cost-effective injection molding process.

For example, Feng et al. [62] studied the automated and robust multi-objective optimization of thin-walled product injection processes using a hybrid radial basis function-multi-objective genetic algorithm (RBF-MOGA) approach. An ANN was used to map the decision variables (e.g., melt temperature, injection time, cooling time, mold temperature, and packing pressure profile) as a function of the objectives to minimize (e.g., warpage, shrinkage, and weld lines). Multiple objectives were articulated by optimizing the Pareto front using the RBF-MOGA approach, achieving a balance between the objectives.

## 3. The Literature on the Optimization of Injection Molding

### 3.1. Organization of This Review

This review is structured into two main sections: (i) optimization of the process as a whole and (ii) optimization specific to the CCCs Given that the methodology used for optimization heavily depends on how the objectives are addressed—whether it involves optimizing a single objective, multiple objectives, or aggregating them into a unified framework—the first section is further subdivided into three categories: single-objective optimization (Table 2), optimization by aggregating multiple objectives into a single function (Table 3), and optimization using multi-objective algorithms (Table 4). The second section, which focuses on the CCCs, is summarized in Table 5.

Table 2 provides a comprehensive overview of single-objective optimization approaches, detailing essential aspects such as the optimization method employed, the specific step within the IM process, the type of decision variables (DVs) considered, the objectives addressed, the modeling techniques applied, the use of surrogate models, and relevant references.

Table 3, Table 4 and Table 5 examine multi-objective optimization scenarios. They differ in structure by replacing the column dedicated to objectives with the number of objectives tackled. This distinction highlights the complexity and diversity of approaches used in these cases.

These tables collectively capture the most critical elements that need to be analyzed to understand how the optimization of the IM process is typically approached in the literature. This review aims to provide a clear and detailed framework for researchers and practitioners by categorizing and organizing the methodologies based on their objectives and techniques. The insights gained here serve as a foundation for advancing optimization strategies, particularly by identifying trends, gaps, and opportunities in the current body of work.

### 3.2. Global Process Optimization

#### 3.2.1. Single-Objective Optimization

Table 2 provides a comprehensive overview of single-objective optimization techniques applied to the injection molding process. It is organized systematically to facilitate understanding of the methodologies, steps, and tools used. The order of the columns reflects a logical progression, starting with the optimization method and ending with references to the relevant studies. This explanation critically evaluates the table by focusing on its structure and referencing only the objectives listed in the fourth column.

##### Optimization Method

This column categorizes the various optimization techniques employed, ranging from traditional approaches, such as empirical methods and Design of Experiments (DOE), to modern techniques, including Particle Swarm Optimization (PSO) and Machine Learning (ML). Hybrid methods, such as EA + PSO, highlight the increasing complexity and adaptability of optimization strategies.

##### Step in Injection Molding

The second column specifies the stage of the injection molding process targeted for optimization. Stages like filling, cooling, and the entire process are noted, reflecting the broad scope of optimization efforts. For instance, cooling steps often focus on objectives like temperature uniformity, while entire-process optimizations target warpages and other defects.

##### Type of Decision Variables (DVs)

The decision variables—whether operational conditions (OC), machine controls (MC), or geometric variables (GL)—define the parameters adjusted during optimization. This column provides insight into the control mechanisms to achieve the desired objectives.

##### Optimization Objectives

The optimization objectives are the primary focus of this critique. These objectives include minimizing defects, warpage, pressure, and weight. Some notable trends and insights include:
1.Warpage: This is the most commonly optimized objective, appearing in numerous studies across various methods. For example:
Ogawa et al. [63] optimized defects using an empirical method.Nguyen et al. [64] utilized the Taguchi method to minimize warpage.Li et al. [65] applied PSO for warpage optimization, showcasing the adoption of modern algorithms.
2.Defects and Surface Quality: Objectives like surface defects [66] and blush defects [67] improve product esthetics and functionality.3.Mechanical Properties: Von Mises stress is another significant objective addressed by methods like PSO [68] and EA [69]. These efforts underline the importance of structural integrity in injection-molded components.4.Weight Optimization: Studies like [70] have shown that optimizing part weight is crucial for applications requiring lightweight yet durable components.

##### Modeling Approach

The modeling tools employed, such as Moldflow, Moldex3D, and experimental setups, play a critical role in analyzing and predicting the outcomes of optimization strategies. Their application varies depending on the complexity and objectives of the optimization.

##### Surrogate Models

Surrogate models like artificial neural networks (ANNs), Kriging, and XGBoost are frequently used to approximate objective functions, especially for computationally expensive problems. For instance:
ANN was employed by Dejene and Wolla [71] for overflow optimization.Kriging was used by Gao and Wang [72] to minimize warpage.Advanced machine learning models like XGBoost [73] demonstrate the evolving landscape of optimization techniques.

##### References

The references provide a basis for tracing the methodologies and verifying the findings, adding credibility to the table.

The optimization objectives, listed in the fourth column, provide a focal point for evaluating the effectiveness and adaptability of the optimization methods. By referencing the objectives, a critical assessment of the table reveals patterns and insights:1.Common Objectives: Warpage emerges as the most frequently optimized objective, with studies spanning diverse methodologies, such as empirical approaches [63], Taguchi methods [64], and advanced evolutionary algorithms [65]. This prevalence underscores warpage as a persistent challenge in injection molding.2.Diversity in Optimization Goals: Objectives vary significantly, from part weight [70] and pressure [74] to defect reduction [67]. This diversity highlights the broad scope of optimization in improving product quality and manufacturing efficiency.3.Advanced Computational Techniques for Complex Objectives: The use of surrogate models like ANN, Kriging, and XGBoost for objectives such as von Mises stress [68,73,75] reflects the trend of leveraging machine learning and computational intelligence to tackle intricate, multivariable problems.4.Evolving Trends: The recent inclusion of ML-based approaches, particularly for objectives like weld line optimization [76] illustrates a shift toward integrating machine learning into traditional optimization workflows. Integration of Hybrid Methods: The combination of EA and PSO [67] for reducing blush defects showcases the potential of hybrid methodologies in addressing complex optimization objectives.

Table 2, through its structured organization, provides an insightful snapshot of the advancements concerning optimizing the injection molding process. Focusing on the optimization objectives reveals the persistent challenges, evolving strategies, and increasing reliance on advanced computational tools and machine learning techniques to achieve better process outcomes. This highlights the dynamic nature of research in this field and sets the stage for further exploration of multi-objective optimization approaches.

The study of Baruffa et al. [76] is a good example of integrating modeling and experimental data to optimize a qualitative objective, specifically the presence or absence of surface defects. This study employs a transfer learning-based artificial neural network (ANN) that utilizes both simulated and experimental data to predict weld line visibility in injection molding processes. Figure 6 illustrates that the ANN is initially trained using molding trials and simulation data, and is subsequently fine-tuned using only molding trials. The research demonstrates that transfer learning can reduce the data required for model training while maintaining predictive accuracy.

#### 3.2.2. Multi-Objective Optimization Using Aggregation Methods

Injection molding is a complex manufacturing process, and its optimization often involves multiple conflicting objectives. Table 3 presents a detailed comparison of studies that employ aggregated multi-objective optimization techniques for injection molding. This table is organized in a logical sequence, beginning with the optimization method and followed by the injection molding process under study. Subsequent columns outline the type of decision variables, number of objectives, modeling tools, surrogate models used, and references. This systematic layout enables a comprehensive assessment of the aggregation techniques used in various studies. Table 3 is similar to Table 2 except in column 4, where the number of objectives replaces the objectives.

##### Optimization Method

Concerning optimization methodologies (column 1), the following observation can be made:
Empirical methods dominate optimization studies, frequently using OC as decision variables to simulate and refine process conditions.Gradient-based methods and Taguchi designs are versatile and effectively capture the interactions of OCs with other decision variables, such as cooling channels or gate locations.Nature-inspired algorithms, such as PSO and EA, excel in handling multiple objectives, often incorporating OCs with other design variables.

**Table 2 polymers-17-00919-t002:** Optimization using a single objective (DOE—design of experiments; GRA—gray relational analysis; SQP—sequential quadratic programming; PSO—particle swarm optimization, EA—evolutionary algorithm; ML—machine learning; OCs—operating conditions; MCG—mold cavity geometry; CC—cooling channel; GL—gate location; RG—runners geometry; PG—part geometry; ANN—artificial neural network; QRS—quadratic response surface; RSM—response surface methodology; GPR—gaussian process regression; XGBoost—extreme gradient boosting).

Optimization	Step	Type DVs	Objectives	Modeling	Surrogates	Reference
Empirical	Entire	OC	Defects	Experimental	ANN	Ogawa et al. [63]
Empirical	Post-Ejec	OC + MCG	Warpage	Moldex3D	--	Kastelic et al. [77]
Complex	Entire	PG	Warpage	Experimental	--	Lee and Kim [78]
DOE	Cooling	CC	Temperature	Moldflow	QRS	Rhee et al. [79]
DOE	Fill + Pack.	OC	Shrinkage	Moldflow	Metamodeling	Villarreal-Marroquin et al. [80]
Taguchi	Entire	OC	Warpage	Moldex3D	RSM	Nguyen et al. [64]
Taguchi	Entire	OC	Overflow	Experimental	ANN	Dejene and Wolla [71]
Tag. + GRA	Entire	OC	Warpage	Moldflow	--	Lin et al. [81]
SQP	Filling	RG	Surface def.	Moldflow	Kriging	Ma et al. [66]
Other	Entire	OC	Warpage	Moldflow	Kriging	Gao and Wang [72]
Other	Entire	OC	Warpage	Moldflow	QRS	Zhang et al. [82]
Other	Entire	OC	Warpage	Moldflow	GPR	Xia et al. [83]
Other	Entire	OC	Von Mises	M + A	Kriging	Liu et al. [75]
Other	Filling	GL	Pressure	Moldex3D	Kriging	Hsu et al. [84]
Other	Fill + Pack.	OC	Pressure	Moldflow	Various	Saad et al. [74]
PSO	Entire	OC	Von Mises	Moldflow	ANN	Xu et al. [68]
PSO	Entire	OC	Warpage	Moldflow	Kriging	Li et al. [65]
EA	Entire	OC	Sink marks	Not specified	ANN	Shen et al. [85]
EA	Cooling	GL	Von Mises	Moldflow	--	Kurkin et al. [69]
EA	Entire	OC	Weight	Moldex3D	XGBoost	Ma et al. [73]
EA + PSO	Entire	RG + OC	Blush defect	Moldflow	ANN	Ardestani et al. [67]
ML	Entire	OC	Part weight	Cadmould	ANN	Lockner and Hopmann [70], Lockner et al. [86]
ML	Entire	OC	Weld line	Moldex3D	ANN	Baruffa et al. [76], Pieressa et al. [87]

##### Step in Injection Molding

This column identifies the optimized steps, which are very similar to those used in the previous case.

##### Type of Decision Variables (DVs)

The third column of Table 2 lists the decision variables considered in the optimization studies. These variables have a significant impact on the quality and efficiency of the injection molding process. The key DV types include the following:1.OCs (Operating Conditions): Operating conditions are the most frequently studied decision variable, underscoring their critical role in injection molding optimization. These variables typically include process parameters such as temperature, pressure, injection speed, and cooling time. Optimizing OCs is fundamental to improving product quality, reducing defects, and enhancing process efficiency. For instance: Fu and Ma optimized operating conditions during the ejection stage to reduce defects like warpage [88], and Nasir et al. employed RSM to refine operating conditions during the cooling and packing phases, achieving improved dimensional accuracy [89]. The widespread focus on OCs directly influences the material behavior during molding and the resulting part quality.2.CC (Cooling Channel): Decision variables related to cooling channels are vital for optimizing the cooling stage of injection molding. Studies such as Cervantes-Vallejo et al. focus on designing efficient cooling systems to achieve uniform temperature distribution and minimize cycle time [90].3.GL (Gate Location): Gate location decision variables are crucial for optimizing the filling phase. Proper gate placement helps enhance material flow, reduce stress concentrations, and minimize defects like weld lines. Li and Wang demonstrated the importance of gate location optimization for achieving better mechanical properties and filling efficiency [91].4.RG (Runner Geometry): Runner geometry decision variables aim to optimize the distribution of molten material across the mold cavities. Efficient runner designs minimize material wastage and pressure loss, creating a balanced filling process.5.PG (Part Geometry): Decision variables related to part geometry are fundamental for optimizing manufacturability and product performance. Park et al. considered part geometry in their optimization, highlighting its role in reducing material usage while maintaining structural integrity [92].

##### Number of Objectives

The number of objectives ranges between 2 and 5, but most works use 2 or 3 objectives. Most objectives are related to minimizing warpage, shrinkage, cycle time, and energy consumption.

For example, the work of Heidari et al. is a typical example with two objectives: minimizing shrinkage and warpage [93,94]. Sreedharan et al. use five objectives based on experimental data, including weld line width after molding and plating, chrome, nickel, and copper plating thickness, length after plating, warpage before and after plating, and step (voltage drop during plating) [95].

##### Modeling Approach

Similarly to the previous case, the use of Moldflow and Moldex3D prevails.

##### Surrogate Models

The last column highlights the aggregation approaches, which play a pivotal role in combining multiple objectives into a single metric or comparable format. Here are the critical insights:1.ANOVA: Fonseca et al., utilized ANOVA to analyze the variance in aggregated objectives, showcasing its effectiveness in determining the impact of key parameters [96].2.GRA (Gray Relational Analysis): Li et al., employed GRA for multi-objective optimization, demonstrating its ability to rank and compare alternatives in complex decision-making scenarios [97].3.Other Aggregation Methods: Fuzzy systems, PCA–GRA combinations, and ANN-based approaches underscore the growing application of computational intelligence to address complex, multi-objective problems with aggregation.

Operating conditions are crucial to injection molding optimization, as they directly influence critical process outcomes, including cycle time, part quality, and energy consumption. Studies focusing on OCs often emphasize trade-offs between competing objectives, such as minimizing warpage while reducing cycle time. These parameters are relatively easy to adjust during process setup, making them a practical and impactful focus for optimization.

Aggregation techniques such as ANOVA, GRA, and hybrid approaches provide robust frameworks for addressing the complexities of multi-objective optimization. Future research should continue to explore advanced aggregation methods, particularly those that integrate machine learning and computational intelligence, to refine decision-making and enhance the efficiency of injection molding processes.

#### 3.2.3. Multi-Objective Optimization

Table 3 presents a detailed overview of multi-objective optimization (MOO) approaches applied to the injection molding process. It organizes information into several columns: Optimization Step, Type, DVs (Design Variables), N.Objs (Number of Objectives), Modeling, Surrogates, and References. Each column’s organization provides insight into the specific focus and advancements in optimizing this process. Below is a critical analysis of the table’s structure and content, starting with a column-by-column analysis:

##### Optimization Method

The first column identifies the optimization methods used. It is fascinating to note the evolution of these methods. Early studies employed more straightforward methods like SAO [98] and basic metaheuristics (e.g., PSO in [99]). However, over time, advanced methods like MOEA/D [100] and NSGA-III [101] have become more prevalent, enabling solutions to more complex, real-world problems. This evolution keeps us at the forefront of the latest trends in the field.

##### Step in Injection Molding

The second column categorizes the optimized injection molding process stage: Filling, Packing, Cooling, or the Entire process. Optimization dominance across the Entire process highlights the trend toward holistic solutions [101,102,103]. This aligns with the increasing industrial demand for comprehensive performance improvement. Isolated stages like Cooling are less represented, likely because standalone optimizations may not sufficiently address interdependent variables across the process [7].

**Table 3 polymers-17-00919-t003:** Optimization using aggregation functions (1—hybrid Taguchi–WASPAS—Ant Lion Optimization; DOE—design of experiments; GRA—gray relational analysis; SQP—sequential quadratic programming; PSO—particle swarm optimization, EA—evolutionary algorithm; OCs—operating conditions; CC—cooling channel; GL—gate location; GG—gate geometry; PG—part geometry; RBF—radial basis function; ANN—artificial neural network; RSM—response surface methodology; GPR—gaussian process regression; PR—polynomial regression; FSA—fast strip analysis).

Optimization	Step	Type DVs	N. Objs	Modeling	Surrogates	Reference
Empirical	Entire	OC	3	Moldflow	--	Wang et al. [104]
Empirical	Ejection	OC	3	Moldflow	--	Fu and Ma [88]
Empirical	Cooling	GL + OC + CC	3	Moldflow	--	Al-Hadad and Wang [105]
Empirical	Cool. + Pack.	OC	2	Moldflow	RSM	Nasir et al. [89]
Empirical	Entire	OC	2	Moldflow	RSM	Meiabadi et al. [106]
Empirical	Entire	OC + GL	4	Custom	--	Trinh [49]
Simplex	Entire	OC	3	Moldflow	--	Sherbelis at al. [22]
Gradient	Entire	OC + GL	3	Moldflow	Kriging	Li and Wang [91]
Gradient	Entire	OC	2	Moldflow	RSM, RBF	Heidari et al. [93,94]
Gradient	Entire	OC	2	Moldflow	--	Hiyane-Nashiro et al. [107]
DOE	Entire	OC	3	Moldflow	Regression	Rodríguez-Yáñez et al. [108]
DOE	Entire	OC	2	Moldex3D	--	Huang et al. [11]
Taguchi	Entire	OC	2	Experimental	RSM	Lan et al. [109]
Taguchi	Entire	OC	2	Moldflow	RSM	Ryu et al. [110]
Taguchi	Entire	OC	3	Moldflow	--	Idayu et al. [111]
Taguchi	Entire	OC	2	Moldflow	--	Ashaari and Amin [112]
Taguchi	Fill. + Pack.	OC	3	Moldex3D	--	Vasiliki et al. [113]
Taguchi	Fill. + Cool.	OC	3	Moldflow	ANOVA, GRA	Md Ali et al. [114]
Taguchi	Entire	OC	2	Moldflow	GRA	Wu et al. [115]
Taguchi	Entire	OC	2	Moldex3D	GRA	Huang et al. [116]
Taguchi	Entire	OC	3	Moldflow	GRA	Li et al. [97]
Taguchi (1)	Entire	OC	2	Experimental	--	Ravikiran et al. [117]
RSM	Entire	OC + GG	5	Experimental	PCA—GRA	Sreedharan et al. [95]
SQP	Entire	PG	3	Moldex3D	PR	Park et al. [92]
Other	Entire	OC	2	Moldflow	GPR	Villarreal-Marroquín et al. [118]
Other	Cooling	OC	2	Moldex3D	RBF	Chang et al. [119]
Other	Entire	OC + PG	4	Moldlow	ANOVA	Fonseca et al. [96]
Fuzzy	Entire	OC	5	Experimental	SQP	Tan and Yuen [120]
PSO	Fill.+ Cool.	OC	3	Custom	FSA	Zhao et al. [48]
PSO	Entire	OC	3	Experimental	ANN	Bensingh et al. [121]
PSO	Entire	OC	3	Moldflow	RSM	Roslan et al. [122]
PSO	Entire	OC	3	Moldflow	ANN	Lin et al. [123]
PSO	Cooling	CC	6	Moldex3D	RSM	Cervantes-Vallejo et al. [90]
EA	Entire	OC	2	Moldflow	--	Deng et al. [124]
EA	Cooling	OC	2	Moldflow	--	Chen et al. [125]
EA	Entire	OC	2	Moldflow	--	Natalini et al. [126]
EA, SQP	Entire	OC	2, 3	Experimental	Kriging	Mukras et al. [127], Mukras [44]
EA	Entire	OC	2	Moldflow	ANN + SVM	Song et al. [128]
EA	Entire	OC	2	Moldflow	ANN	Yang et al. [129]
EA, EA-PSO	Entire	OC	2	Experimental	ANN, RSM	Nguyen et al. [130], Chen et al. [131]

##### Type of Decision Variables (DVs)

In this case (column 3), most studies focus on optimizing operating conditions [62,132] reflecting the centrality of temperature, pressure, and timing in ensuring product quality. Combinations like operating conditions and gate geometry [133] and operating conditions and gate location [134] underscore efforts to tackle coupled variables. These studies address challenges such as balancing material flow and structural integrity, which are crucial for achieving optimal designs. The dominance of OCs as the primary design variable across studies [135,136] demonstrates the emphasis on core controllable parameters in injection molding. Less common variables like glass fiber content [137], represent innovative attempts to expand the design space, addressing specialized requirements such as lightweight and reinforcement.

However, some challenges remain. While the operating conditions dominate as decision variables, combinations like operating conditions with gate geometry GG [133] or with glass fiber content [137] remain underexplored. These areas present significant opportunities for future research to address specific challenges in advanced injection molding scenarios. The urgency and importance of these research areas cannot be overstated; they should be a key focus of our future work.

##### Number of Objectives

Most optimizations target 2 to 3 objectives [138,139] balancing complexity and computational feasibility. Typical objectives include minimizing warpage, cycle time, and energy consumption. Studies tackling higher-dimensional objectives, such as NSGA-III with seven objectives [101], signify advances in managing many-objective optimization, leveraging algorithms designed for complex trade-offs. However, dealing with more than three objectives can be challenging [140,141,142].

##### Modelings

Simulation tools such as Moldflow and Moldex3D dominate, and studies like those by Cheng et al. [143] and Kitayama et al. [144] demonstrate their widespread. These tools offer reliable process simulation capabilities, reducing dependency on costly experimental setups. Experimental approaches [135,145] are rare, reflecting the higher costs and challenges associated with physical testing compared to simulation.

##### Surrogate Methods

The extensive use of surrogate models like ANN [100,137] and Kriging [138,146] highlights the need for computational efficiency in solving expensive simulation-based optimization problems. Bayesian approaches [136] and Gaussian Process Regression [147] are less common but represent promising methods for incorporating uncertainty in optimization models. Surrogate models such as ANN [137,148] and RSM [149] have gained traction due to their ability to approximate expensive simulations effectively. Studies by Zhao and Li [150] and Zeng et al. [136] highlight emerging interest in combining surrogate modeling with uncertainty quantification.

##### References

The progression in references from foundational works [151,152] to more recent studies [102,153] illustrates the field’s iterative advancements. Kitayama et al. [98,144,154,155,156,157,158,159,160,161,162] have contributed extensively, particularly in integrating surrogate modeling with simulation-based optimization, underscoring their role in advancing the field. Later studies, such as those by Zhai et al. [163] and Guo et al. [137] explore newer methodologies and higher-dimensional optimization problems, showing the field’s growing sophistication.

With extensive contributions from key researchers, such as Kitayama et al. [98,144,154,155,156,157,158,159,160,161,162], Fernandes et al. [15,134,152,164,165,166], and Zeng et al. [136], the field demonstrates significant progress in addressing complex, real-world challenges. Future directions should focus on:

Expanding the range of design variables, including unconventional combinations.Addressing computational challenges in many-objective optimization.Incorporating experimental validation to complement simulation studies.

By leveraging the lessons from this body of research, the optimization of injection molding can continue to evolve, enabling more efficient, cost-effective, and high-quality manufacturing processes.

### 3.3. Optimization of CCCs

Before delving into the optimization and design of conformal cooling channels (CCCs), it is essential to review foundational studies on their modeling and assessment.

Numerous investigations have demonstrated the advantages of CCCs in injection molding, particularly in improving cooling efficiency, reducing cycle time, and enhancing part quality. For instance, Brooks and Brigden [167] designed and validated conformal cooling layers with self-supporting lattices for additively manufactured tooling, showing significant reductions in cooling time, warpage, and sink marks through empirical testing. Similarly, Mazur et al. [168] Combined numerical and experimental approaches are used to assess the cooling efficiency and part deformation of CCCs produced via Selective Laser Melting (SLM).

Van As et al. [169] compared tool steel inserts fabricated by Direct Metal Laser Sintering (DMLS) with those made using traditional manufacturing techniques, highlighting improvements in cycle time and cost efficiency. Tuteski and Kočov [170] further examined the effectiveness of CCCs over conventional cooling systems, emphasizing their benefits in terms of cycle time, temperature uniformity, and cooling performance.

More recently, advanced methodologies have been applied to CCC research. Konuskan et al. [171] leveraged machine learning, specifically LSTM neural networks, to predict cooling profiles, optimizing cooling times through Moldflow modeling. Zacharski et al. [172] explored modular injection molds with CCCs, demonstrating design flexibility and cost savings without compromising cooling efficiency.

These studies collectively illustrate the progression of CCC research, from empirical evaluations to the integration of machine learning techniques, underscoring the transformative impact of CCCs on injection molding technology. However, recent advancements in additive manufacturing decisively influenced the design of CCCs, allowing the development of different technical designs that impact the cooling system’s performance by reducing the cycle time while reducing potential defects [8,9,16,18].

Table 4 provides a comprehensive summary of optimization methods used in designing conformal cooling channels (CCCs) for the injection molding process. While it contains information similar to that presented in earlier tables, the main difference lies in the arrangement of the columns. Unlike the previous tables, which presented each type of method used to address the objectives separately, this table consolidates all references into a single, unified format.

The studies reviewed emphasize the critical role of optimization in improving the performance of CCCs, which directly impacts the efficiency, quality, and sustainability of injection molding operations. Researchers have advanced the state of CCC design by carefully selecting and refining design variables, modeling tools, and optimization techniques; however, significant opportunities remain to enhance its effectiveness further. Optimization addresses immediate operational challenges and contributes to long-term advancements in manufacturing technology.

The optimization of CCCs is crucial because these channels significantly influence the thermal management of molds, affecting cooling time, dimensional stability, and product quality. Without optimized CCC designs, manufacturers face inefficiencies, such as longer cycle times, higher energy consumption, and defective parts. Studies, such as those by Mohd Hanid et al. [173] and Shayfull et al. [174], demonstrate how optimizing operative conditions (OCs) and CCC geometries enhances cooling performance and reduces production costs. Moreover, the incorporation of refined cooling strategies has shown the potential to extend mold life and improve the overall sustainability of the process.

Optimization techniques, such as evolutionary algorithms (EAs), as seen in Mercado-Colmenero et al. [175,176] and empirical methods, as utilized by Dimla et al. [177] and Chaabene et al. [178] have been extensively used. However, the predominance of single-objective (SO) optimization across many studies suggests a limited exploration of the full potential of CCC design. Addressing these limitations could unlock more efficient designs that simultaneously optimize multiple performance metrics.

The need for multi-objective optimization (MO) is increasingly evident. MO approaches allow researchers to address trade-offs between competing objectives, such as minimizing cooling time, ensuring uniform temperature distribution, and maintaining mold structural integrity. While studies like Kanbur et al. [179] and Le Goff et al. [180] explore MO techniques, they represent only a tiny fraction of the literature. These techniques enable a more nuanced understanding of the relationships among conflicting objectives, such as balancing cost and performance.

Choosing the right objectives is equally critical. While cooling efficiency and cycle time are commonly targeted, newer studies, such as Wu and Tovar [41] have integrated structural and thermal considerations using topology optimization (TO). Future work must consider incorporating objectives that address sustainability, material usage, and cost, ensuring a holistic approach to CCC optimization. Additionally, optimization methods should incorporate robustness analysis to account for variations in material properties and manufacturing conditions, which are often overlooked in single-objective studies.

Topology optimization (TO) has emerged as a powerful tool for achieving optimal material distribution while maintaining structural and thermal performance. Studies like Jahan et al. [181,182] highlight how TO combined with thermo-mechanical modeling can refine CCC designs for durability and heat dissipation. Wu and Tovar [41] demonstrate that TO integrates seamlessly into gradient-based optimization methods, enabling more refined control over complex geometries. Such integration has led to significant improvements in mold performance and cooling efficiency.

TO is particularly valuable in generative design (GD), as explored by Wilson et al. [13]. Combining TO and GD can lead to innovative CCC configurations that outperform traditional designs. These techniques enable the exploration of unconventional geometries that enhance thermal performance while minimizing material usage. Future research should focus on integrating TO with real-time simulation and advanced optimization frameworks to further expand its application. Furthermore, combining TO with machine learning models can enhance the design process by predicting optimal configurations based on historical data and real-time feedback.

**Table 4 polymers-17-00919-t004:** Optimization using multi-objective optimization algorithms ((1)—Glass fiber content; (2)—Plasticating; M + Ab—Moldlow + ABACUS; SAO—sequential approximate optimization; SD—sequential design; MBO—multi-objective Bayesian optimization; MOFA—multi-objective firefly algorithm; PSO—particle swarm optimization; MPSO—multi-objective PSO; SQP—sequential quadratic programming; MPDE—multi-population differential evolution; MOEA—multi-objective evolutionary algorithm; OCs—operating conditions; GG—gate geometry; GL—gate location; RG—runners geometry; PG—part geometry; RBF—radial basis function; ANN—artificial neural network; RSM—response surface methodology; GPR—gaussian process regression).

Optimization	Step	Type DVs	N.Objs	Modeling	Surrogates	Reference
SAO	Pack. + Cool. Cooling, Entire, Fill. + Pack.	OC	2, 3	Moldex3D	RBF	Kitayama et al. [98,144,154,155,156,157,158,159,160,161,162,163]
SAO	Fill. + Pack.	OC	2	Moldex3D	RBF	Hashimoto et al. [183]
SD	Cooling	OC	3	Moldex3D	GPR	Chen et al. [7]
MBO	Entire	OC	2	Moldflow	GPR + ANN	Jung et al. [132]
MOFA	Entire	OC (1)	2	Moldflow	ANN	Liu et al. [148]
MOPSO	Entire	OC	3	Moldflow	ANN	Zhang et al. [102]
MOPSO	Entire	OC	3	Moldflow		Liu et al. [21]
PSO	Entire	OC	2	Moldflow	Kriging	Chen et al. [99]
PSO + SQP	Entire	OC	2	CATIA		Mehta and Padhi [153]
MPDE	Entire	OC + GG	3	Moldflow	Kriging	Wang et al. [133]
MOEA	Fill. + Pack.	OC/ OC + GL	3, 5	Moldflow		Fernandes et al. [134,152,164,165,166]
MOEA	Entire	OC	3	Moldflow	ANN	Feng et al. [62,184]
MOEA	Entire	OC	3	M + Ab	ANN	Fonseca et al. [185]
MOEA/D	Entire	OC + GG	2	Moldflow	ANN	C. Wang et al. [100]
NSGA-II	Filling	RG + OC	3	Moldflow		Alam and Kamal [151]
NSGA-II	Filling	RG + OC	5	Moldflow		Zhai et al. [186]
NSGA-II	Entire	OC + RG Geometry	3	Moldflow		Ferreira et al. [187,188]
NSGA-II	Entire	RG + OC	3	Moldflow	ANN	Cheng at al. [143]
NSGA-II	Entire	OC	2	Moldflow	RSM	Park and Nguyen [146]
NSGA-II	Entire	OC	2	Moldflow	Kriging	Zhao and G. Cheng [138]
NSGA-II	Entire	OC	4	Moldflow	RSM	Tian et al. [103]
NSGA-II	Entire	OC	3	Moldflow	RSM	Li et al. [189]
NSGA-II	Entire	PG + OC	2	Moldflow	RSM	Zhijun et al. [190]
NSGA-II	Entire	OC	2	Moldflow	ANN	Lu and Huang [139]
NSGA-II	Entire	OC	2	Experimental	ANN	Wang et al. [135]
NSGA-II	Entire	OC (1)	2	Moldflow	RSM	Zhao and K. Li [150]
NSGA-II	Entire	OC	3	Moldex3D	ANN	Zhai et al. [163]
NSGA-II	Entire	OC	2	Experimental	Kriging	Chang et al. [145]
NSGA-II	Entire	OC	3	Moldflow	ANN	Guo et al. [137]
NSGA-II	Entire	OC + Gas	3	Moldflow	ANN	Guo et al. [137]
NSGA-II	Entire	OC	3	Moldflow	Bayesian	Zeng et al. [136]
NSGA-II	Entire (2)	OC	2	Moldflow	GPR	Kariminejad et al. [147]
NSGA-III	Entire	OC	7	Moldex3D	ANN	Alvarado-Iniesta et al. [101]
NSGA-III	Entire	OC	4	Moldflow	RSM	Zhou et al. [149]
NSGA-III	Entire	OC	4	Moldflow	RSM	Zhu et al. [191]

Several studies in Table 5 provide critical insights into the optimization process. For instance:
Wang et al. [192] utilized Kriging-based surrogate modeling to optimize CCCs and gate geometry, demonstrating how surrogate models can reduce computational costs while maintaining accuracy.Silva and Rodrigues [193] and Kanbur et al. [194] leveraged advanced surrogate and machine learning methods like ANN, showing the potential of AI-driven optimization for complex geometries. These studies highlight the importance of integrating AI techniques with traditional simulation tools to achieve enhanced performance.Empirical methods as employed by Hsu et al. [195] and Saifullah et al. [196] remain popular but highlight the limitations of relying on trial-and-error approaches to achieve globally optimal designs. Expanding on these methods with advanced computational tools could yield more robust and adaptable solutions.

While significant progress has been made, several gaps and opportunities remain in the optimization of CCCs:1.Expanded Multi-Objective Frameworks: Future research should adopt more robust multi-objective optimization techniques, such as NSGA-II, NSGA-III, or MOEA/D, to capture and address cooling performance, cost, and sustainability trade-offs. Objectives should be carefully selected to reflect real-world constraints and priorities. Multi-objective studies should also incorporate advanced visualization techniques to enable better decision-making.2.Integration of Advanced Modeling Tools: Combining simulation tools like Moldflow, ANSYS, and COMSOL with experimental validation will improve the reliability of optimization results. Hybrid frameworks, as demonstrated by Jahan et al. [197] and Shen et al. [198] are particularly promising. Integrating cloud computing and parallel processing could further enhance the scalability of these hybrid frameworks.3.Sustainability and Cost Optimization: The increasing focus on green manufacturing necessitates incorporating life cycle analysis (LCA) into CCC design. Future studies should optimize for material efficiency and energy savings alongside thermal performance. Incorporating sustainability metrics into optimization frameworks could drive innovation in eco-friendly mold designs.4.AI-Driven Optimization: Machine learning (ML) techniques, as explored by Gao et al. [42], should be further developed for predictive modeling, real-time optimization, and adaptive cooling strategies. Integrating AI with topology optimization could open new possibilities for intelligent and autonomous CCC designs. AI-driven methods could also be used to develop predictive maintenance schedules for molds, enhancing their operational lifespan.5.Generative Design and Dynamic CCCs: GD and DCCCs, as discussed by Wilson et al. [13] and Kirchheim et al. [12], have the potential to revolutionize CCC design. Research should focus on scaling these approaches for industrial applications while addressing computational challenges. Cloud-based generative design platforms could democratize access to these advanced technologies, enabling broader adoption.6.Real-World Applications and Validation: Despite the progress in modeling and simulation, empirical studies like those by Eiamsa-Ard and Wannissorn [199] highlight the importance of real-world testing. Future work should emphasize validation in industrial settings to bridge the gap between theory and practice. Collaborations with industry stakeholders could facilitate the development of CCC designs tailored to specific manufacturing scenarios.7.Exploration of Novel Materials: Advancements in materials science could play a pivotal role in optimizing CCC designs. Future studies should explore using advanced composites and metal alloys to enhance molds’ thermal and mechanical properties. The integration of material-specific optimization methods could lead to breakthroughs in CCC performance.

By addressing these challenges and building on the foundations laid by the works in Table 5, future research can further enhance CCC performance, efficiency, and sustainability in injection molding processes. This evolution will improve manufacturing outcomes and contribute to the broader goals of sustainable and cost-effective production. Integrating advanced computational tools, AI, and real-world validation will be key to achieving these objectives, driving innovation in CCC design for years.

**Table 5 polymers-17-00919-t005:** Conformal cooling channels design (EA—evolutionary algorithm; PSO—particle swarm optimization; EI—expected improvement method; DOE—design of experiments; TM + TO—Thermo-mechanical + topology optimization; SA—simulated annealing; SLP—sequential linear programming; GD—generative design; CCC—conformal cooling channels design; OCs—operative conditions; GG—gate geometry; DCCCs—dynamic conformal cooling channels; Mec—mold mechanical properties; M + A—Moldflow + ANSYS; A + C—ANSYS + COMSOL; M + mF—Moldflow + modeFRONTIER; SO–Ag.—SO-Aggregation; RSM—response surface methodology; ANN—artificial neural network; GRA—gray relational analysis).

Opt.	Type DVs	N. Objs	Modeling	Surrog.	Type	Reference
EA + PSO	OC	1	Moldflow	RSM	SO	Mohd Hanid et al. [174]
EI	CCC + GG	1	M + A	Kriging	SO	Wang et al. [192]
Empirical	CCC	3	M + A		SO	Saifullah et al. [196]
Empirical	CCC	3	Moldex3D		SO	Hsu et al. [195]
Empirical	CCC	3	Exp.		SO	Vojnová [200]
Empirical	CCC	2	Moldflow		SO	Yadegari et al. [201]
Empirical	CCC	3	Moldflow		SO	Venkatesh et al. [202]
Empirical	CCC	2	Moldflow		SO	Li et al. [203]
Empirical	DCCC	3	Moldex3D		SO	Kirchheim et al. [12]
DOE + TM	CCC	2	ANSYS		SO–Ag.	Jahan et al. [197]
EA	CCC	3	M + A		SO–Ag.	Mercado-Colmenero et al. [175,176]
EA	CCC	3	Moldex3D	RSM	SO–Ag.	Wang and Lee. [204]
Empirical	CCC	3	Moldflow		SO–Ag.	Dimla et al. [177]
Empirical	CCC + OC	2	Moldflow		SO–Ag.	Shayfull et al. [174]
Empirical	CCC	3	Exp.		SO–Ag.	Eiamsa-Ard and Wannissorn [199]
Empirical	CCC	4	ANSYS		SO–Ag.	Kanbur et al. [205]
Empirical	CCC	3	Moldex3D		SO–Ag.	Godec et al. [206]
Empirical	CCC	2	ANSYS		SO–Ag.	Silva et al. [207]
SA	CCC	3	ANSYS		SO–Ag.	Silva and Rodrigues [193]
ML	CCC	2	Moldflow	ANN	SO–Ag.	Gao et al. [42]
TM + TO	CCC + Mec	3	ANSYS		SO–Ag.	Jahan et al. [181]
TM + TO	CCC	3	A + CL		SO–Ag.	Jahan et al. [182]
TO + SLP	CCC	2	Custom		SO–Ag.	Li et al. [208]
EA	CCC	3	ANSYS	ANN	SO–Ag.	Kanbur et al. [194]
Empirical	CCC	2	Moldflow		SO–Ag.	Marques et al. [209]
Empirical	CCC	2	Moldflow		SO–Ag.	Kamarudin et al. [210]
Empirical	CCC + OC	2	ANSYS		SO–Ag.	Shen et al. [198]
Empirical	CCC	3	Moldflow		SO–Ag.	Chaabene et al. [178]
Grad(TO)	CCC + OC	3	COMSOL		SO–Ag.	Wu and Tovar [41]
Taguchi	CCC	4	SolidWorks	GRA	SO–Ag.	Simiyu et al. [211]
GD	GDCCC	3	Moldex3D		SO-Ag.	Wilson et al. [13]
MOEA	CCC	3	COMSOL		MO	Kanbur et al. [212]
NSGA-II	CCC + Tc	2	M + mF		MO	le Goff et al. [180]

For example, Jahan et al. optimized molds’ thermal and structural performance [197]. As reported by the authors, the cooling time increases to 19.8 s in the printed steel mold, compared to 13.9 s in the traditional steel case. However, the maximum von Mises stress shows minimal variation and remains within the allowable stress limit. This behavior can be attributed to the significant reduction in thermal conductivity, which drops from 60.5 W/m·K in traditional steel to 13.8 W/m·K in printed steel. Nevertheless, analyzing all design cases using experimentally derived material properties is crucial to ensure optimal design configurations.

The optimization of CCCs requires a coupled thermal and mechanical analysis to ensure efficient cooling performance and structural integrity. As demonstrated, while printed steel molds can significantly alter thermal conductivity—resulting in increased cooling times—the mechanical stress distribution remains within acceptable limits. However, neglecting the interaction between thermal and mechanical behavior can lead to suboptimal designs, potentially compromising either efficiency or durability. Therefore, employing experimentally validated material properties and an integrated multi-physics approach is essential for achieving optimal CCC configurations that balance performance and reliability.

As mentioned earlier, Wilson et al. [13] recently employed generative design (GD) and dynamic CCCs to develop CCCs. The GD process iteratively generates numerous design variants within user-defined constraints and quality metrics. These variants are then shortlisted either by the designer (“artificial selection”) or by computational evaluation (“natural selection”). Users can significantly refine the output by customizing the evaluation criteria and streamlining the design search process.

In this study, the approach was implemented by (i) predefining CCC orientations, (ii) restricting CCC diameters based on operating pressures and profile conformity, and (iii) customizing postprocessing of the results. Figure 7 illustrates examples of the generated designs using high-coolant-pressure parallel CCCs, low-coolant-pressure parallel CCCs, perpendicular CCCs, and helical CCCs. All four mold tool design variants are compatible with the part geometry produced by the mold tool. This approach has great potential for designing CCCs.

## 4. Conclusions

Injection molding optimization remains a critical area of research and industrial application due to its profound impact on manufacturing efficiency, product quality, and sustainability. This review underscores the complexity of the injection molding process, with interdependent phases that significantly influence the final product’s quality. Adopting advanced computational tools and optimization methodologies has marked a transformative phase in the industry. Conformal cooling channels (CCCs) represent a breakthrough in cooling efficiency, addressing challenges such as warpage, shrinkage, and reducing cycle time. However, gaps persist, particularly in understanding the causal relationships between decision variables and defects, necessitating further research.

Surrogate models, including Kriging and artificial neural networks, reduce computational costs while maintaining optimization accuracy. Multi-objective optimization algorithms, such as NSGA-II and MOEA, have effectively balanced conflicting objectives, including cooling time, mechanical properties, and material usage. Emerging technologies, including adaptive algorithms and machine learning, offer significant potential for real-time process optimization, enabling the development of more innovative mold designs that adapt to dynamic production requirements.

The selection of optimization objectives plays a decisive role in influencing the convergence of solutions. Objectives such as minimizing warpage, shrinkage, and cycle time often conflict, necessitating advanced methods to balance trade-offs. Techniques such as NSGA-II and MOEA utilize dominance concepts to explore the solution space effectively, while surrogate models like ANN and Kriging help approximate complex simulations, thereby expediting convergence. A precise alignment between the objectives, decision variables, and optimization methodologies ensures robust and practical solutions. These approaches improve computational efficiency and enhance the likelihood of identifying globally optimal designs that meet diverse manufacturing needs.

Specific conclusions from this review include the critical role of cooling phase optimization, which constitutes a substantial portion of the cycle time and directly impacts defects such as warpage and shrinkage. The integration of conformal cooling channels has led to significant advancements in achieving uniform cooling and enhancing product quality. The study also highlights the importance of leveraging computational tools, such as Moldex3D, for simulating complex phenomena, thereby enabling precise decision-making. Furthermore, using surrogate models, such as Kriging and ANNs, to reduce the computational burden of optimization is particularly effective when combined with multi-objective approaches. These findings underscore the need for comprehensive optimization frameworks that incorporate both theoretical and practical insights, ensuring scalability and adaptability in industrial settings.

Future research should focus on integrating sustainability metrics, such as energy efficiency and material reuse, into optimization frameworks. The convergence of artificial intelligence, advanced simulation tools, and robust experimental validations can bridge theoretical and practical gaps, driving future innovations. As manufacturing continues to demand high precision, cost efficiency, and sustainability, the continuous evolution of optimization strategies in injection molding will play a central role in meeting these objectives, contributing to the advancement of global manufacturing technologies.

## Figures and Tables

**Figure 1 polymers-17-00919-f001:**
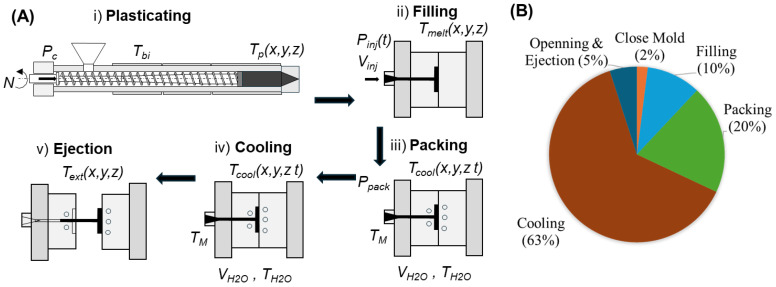
(**A**) The injection molding cycle and (**B**) relative cycle times.

**Figure 2 polymers-17-00919-f002:**
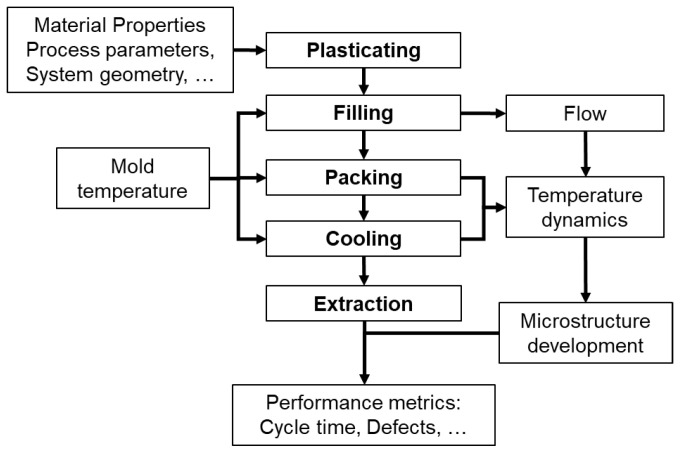
Interrelation between decision variables, cycle phases, and optimization objectives.

**Figure 3 polymers-17-00919-f003:**
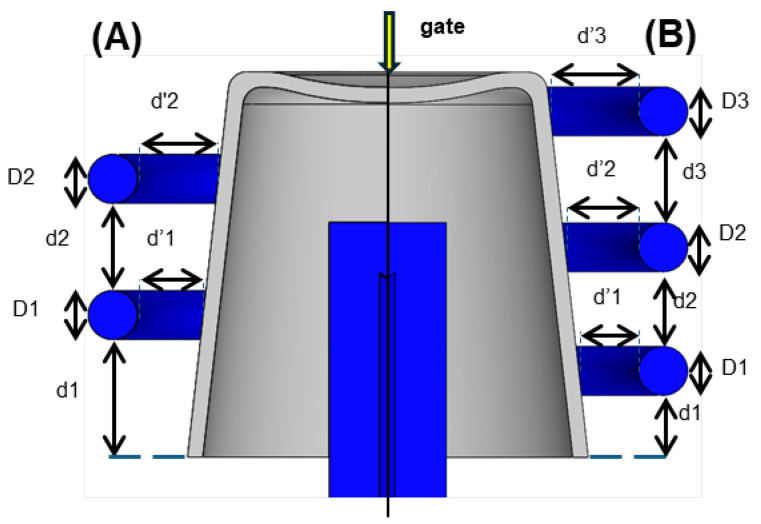
Cooling channels geometry: (**A**) CCC with two circular rings; (**B**) CCC with three circular rings.

**Figure 4 polymers-17-00919-f004:**
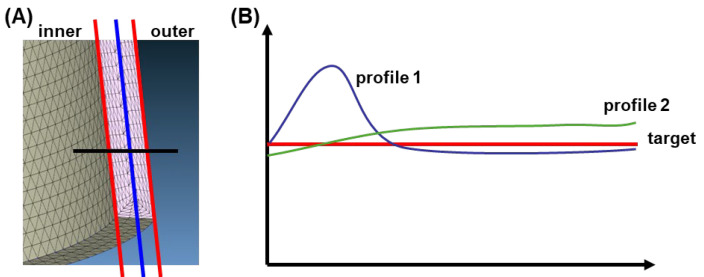
Example of calculation objectives related to temperature: (**A**) identification of the part surfaces; (**B**) temperature profile difference.

**Figure 5 polymers-17-00919-f005:**
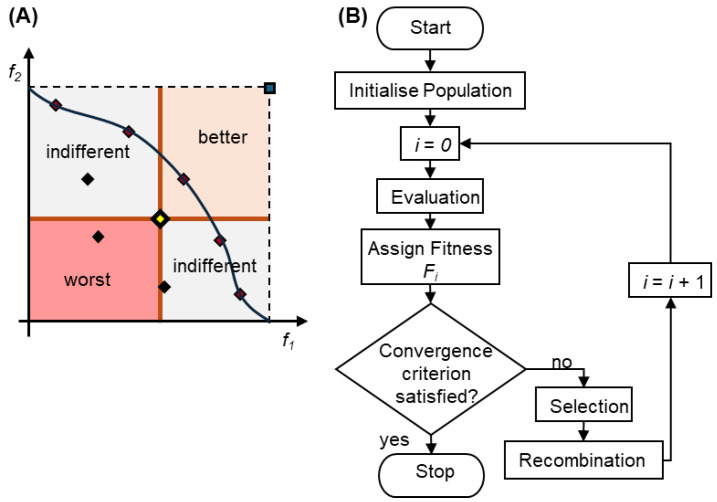
Multi-objective optimization: (**A**) concept of non-dominance and (**B**) MOEA flow chart.

**Figure 6 polymers-17-00919-f006:**
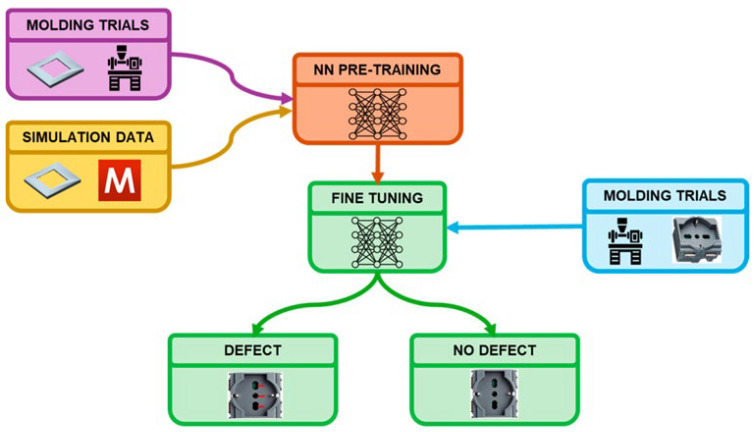
Flowchart of the methodology implemented by Baruffa, et al. [76].

**Figure 7 polymers-17-00919-f007:**
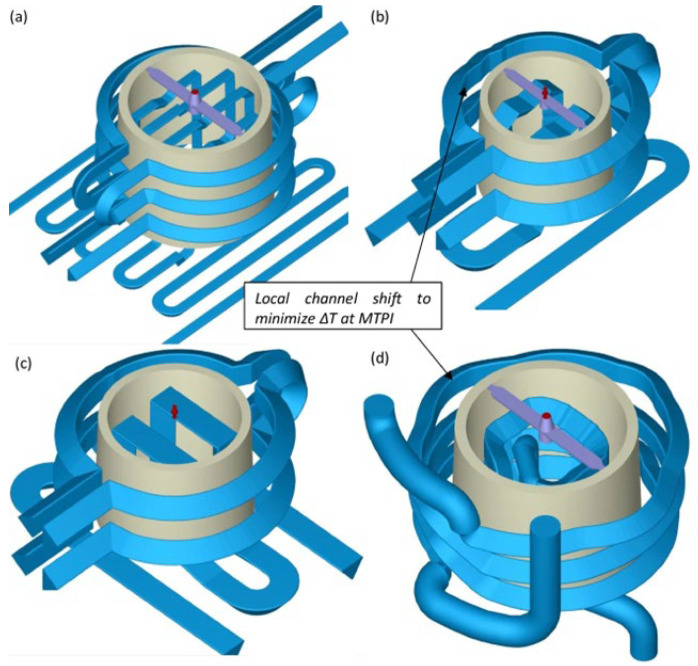
Generated designs using the methodology of Wilson et al. [13]: (**a**) high-coolant-pressure parallel CCCs, (**b**) low-coolant-pressure parallel CCCs, (**c**) perpendicular CCCs, and (**d**) helical CCCs.

## Data Availability

The data supporting this study’s findings are available from the corresponding author upon reasonable request.

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
