# Peer review of "A Review on Injection Molding: Conformal Cooling Channels, Modelling, Surrogate Models and Multi-Objective Optimization"

_polymers, 2025, doi:10.3390/polym17070919_

Round 1
Reviewer 1 Report
Comments and Suggestions for Authors
- The drawbacks associated with the subject under study and the methods required to overcome may be touched upon and are to be mentioned in the Abstract.
- The literature review is to be strengthened in each category to highlight the Research gap.
- In Page 11, it is mentioned as “Non-dominance is key in multi-objective optimization, balancing trade-offs without arbitrary weights. By recombining top solutions, the algorithm evolves a diverse, high-quality approximation of the Pareto front. Please provide literature support for the statement.
- Please provide the literature support for Figure 5. If it is referred from somewhere.
- Please format the Reference section. In some of the References, the title of paper, all letters of each word are of upper case whereas in the other references it is different. Please format all the references accordingly.
Author Response
We thank the reviewers for their valuable comments, which we have incorporated into the manuscript as much as possible. These comments help to improve the text.
Reviewer 1
Q1: The drawbacks associated with the subject under study and the methods required to overcome may be touched upon and are to be mentioned in the Abstract.
ANSWER
The following sentence was added to the abstract, highlighted in yellow in the manuscript: “A significant challenge of this process lies in the need to employ sophisticated computational techniques to optimize the various phases.” Additionally, the drawbacks identified by the reviewer are described in detail in the introduction. See also the answer to the next question.
Also, the English of the abstract was improved.
Q2: The literature review is to be strengthened in each category to highlight the Research gap.
ANSWER
The reviewer is correct; the research gap we intend to cover was not clearly stated. We verified that, in some cases, the most recent works in the literature do not consider the best methods used in previous works that optimize operating conditions, feed systems, or cooling systems, for example. Therefore, to strengthen our idea and to complement what we wrote, we added the following sentence to improve our manuscript:
“An evident trend in the body of research on injection molding optimization is the compartmentalized treatment of distinct objectives, such as the design of conformal cooling channels (CCC), as independent challenges. Despite the availability of advanced methodologies, including multi-objective evolutionary algorithms (MOEAs) and surrogate modeling techniques like artificial neural networks (ANNs), which have demonstrated considerable success in optimizing processing parameters, these approaches are seldom leveraged when addressing new or parallel objectives. Instead, many studies regress to elementary strategies, frequently relying on single-objective formulations and classical design of experiments (DOE), thereby reinitiating the optimization framework without building upon prior methodological progress. This review aims to highlight this fragmentation and advocates for a more coherent and cumulative application of advanced optimization paradigms across the various facets of the injection molding process.
Q3: In Page 11, it is mentioned as “Non-dominance is key in multi-objective optimization, balancing trade-offs without arbitrary weights. By recombining top solutions, the algorithm evolves a diverse, high-quality approximation of the Pareto front. Please provide literature support for the statement.
ANSWER
Two references were added at the end of this sentence: “Deb K (2001) Multi-objective Optimization Using Evolutionary Algorithms. Wiley, Chichester” and “Coello CAC, Lamont GB, Van Veldhuizen DA (2007) Evolutionary Algorithms for Solving Multi-Objective Problems. Springer, Boston, MA”
Q4: Please provide the literature support for Figure 5. If it is referred from somewhere.
ANSWER
Figure 5 is original and was made by the authors.
Q5: Please format the Reference section. In some of the References, the title of paper, all letters of each word are of upper case whereas in the other references it is different. Please format all the references accordingly.
ANSWER
The references were corrected.
Reviewer 2 Report
Comments and Suggestions for Authors
The authors have presented a well-structured and informative review on optimization methodologies in injection molding, particularly emphasizing conformal cooling channels, surrogate modeling, and multi-objective optimization. However, several key areas require clarification and enhancement before acceptance.
- The manuscript should explicitly highlight how this review differentiates itself from existing literature on injection molding optimization. What specific gaps does it address?
- The discussion on CCC is well-articulated, but a comparative analysis of different CCC designs and their impact on cycle time and defect reduction should be expanded. Have recent advancements in additive manufacturing influenced CCC design?
- While various algorithms are discussed, their practical applicability in real-world scenarios needs further elaboration. Are there any case studies demonstrating the effectiveness of these algorithms in industrial settings?
- The review mentions Moldex3D and Autodesk Moldflow but does not compare their capabilities in detail. A comparative table outlining their advantages and limitations would add value.
- The manuscript highlights the use of Kriging, response surface methodology, and artificial neural networks. However, it should discuss the trade-offs between accuracy and computational cost in more depth. How do these models compare in terms of predictive performance and efficiency?
There are minor grammatical inconsistencies and formatting issues that should be corrected.
Author Response
We thank the reviewers for their valuable comments, which we have incorporated into the manuscript as much as possible. These comments help to improve the text.
Reviewer 2
The authors have presented a well-structured and informative review on optimization methodologies in injection molding, particularly emphasizing conformal cooling channels, surrogate modeling, and multi-objective optimization. However, several key areas require clarification and enhancement before acceptance.
Q1: The manuscript should explicitly highlight how this review differentiates itself from existing literature on injection molding optimization. What specific gaps does it address?
ANSWER
The reviewer is correct; the research gap we intend to cover was not clearly stated. We verified that, in some cases, the most recent works in the literature do not consider the best methods used in previous works that optimize operating conditions, feed systems, or cooling systems, for example. Therefore, to strengthen our idea and to complement what we wrote, we added the following sentence to improve our manuscript:
“An evident trend in the body of research on injection molding optimization is the compartmentalized treatment of distinct objectives, such as the design of conformal cooling channels (CCC), as independent challenges. Despite the availability of advanced methodologies, including multi-objective evolutionary algorithms (MOEAs) and surrogate modeling techniques like artificial neural networks (ANNs), which have demonstrated considerable success in optimizing processing parameters, these approaches are seldom leveraged when addressing new or parallel objectives. Instead, many studies regress to elementary strategies, frequently relying on single-objective formulations and classical design of experiments (DOE), thereby reinitiating the optimization framework without building upon prior methodological progress. This review aims to highlight this fragmentation and advocates for a more coherent and cumulative application of advanced optimization paradigms across the various facets of the injection molding process.
Q2: The discussion on CCC is well-articulated, but a comparative analysis of different CCC designs and their impact on cycle time and defect reduction should be expanded. Have recent advancements in additive manufacturing influenced CCC design?
ANSWER
This paper aims not to address specific questions of the IM process, but to present a comprehensive framework for optimizing the process. To accommodate the suggestion, we added the following sentence to the text in section 3.3: “However, recent advancements in additive manufacturing decisively influenced the design of CCC, allowing the development of different technical designs that impact the cooling system's performance by reducing the cycle time while reducing potential defects [Feng, 2021; Kanbur, 2020; Wei 2020; Silva, 2022]”.
Q3: While various algorithms are discussed, their practical applicability in real-world scenarios needs further elaboration. Are there any case studies demonstrating the effectiveness of these algorithms in industrial settings?
ANSWER
These algorithms are applied in the papers cited in the text. Since this is a review paper, the goal is not to demonstrate the effectiveness of the algorithms discussed, but rather to identify where and for what purpose they are applicable. In this way, we expected to help the reader select the best option for the problem they intend to solve. Most of the studies cited in this review tried to demonstrate the effectiveness of the algorithms referred to here in real-world examples. However, it is challenging to determine whether the results were successfully implemented in industry.
Q4: The review mentions Moldex3D and Autodesk Moldflow but does not compare their capabilities in detail. A comparative table outlining their advantages and limitations would add value.
ANSWER
This analysis was already detailed in section 2.6. Table 1 shows the detailed analysis required by the reviewer.
Q5: The manuscript highlights the use of Kriging, response surface methodology, and artificial neural networks. However, it should discuss the trade-offs between accuracy and computational cost in more depth. How do these models compare in terms of predictive performance and efficiency?
ANSWER
The performance of each technique depends on its application, i.e., its data structure. In section 2.6, the analysis required by the reviewer was already made in the original manuscript (page 13). See, for example, the paragraph that finishes with “The effectiveness of each technique depends on the problem’s structure, data availability, and computational resources.”.
Reviewer 3 Report
Comments and Suggestions for Authors
Please see my comments in the attached file. Thanks!

Author Response
We thank the reviewers for their valuable comments, which we have incorporated into the manuscript as much as possible. These comments help to improve the text.
Reviewer 3
Review Report: A Review on Injection Molding: Conformal Cooling Channels, Modelling, Surrogate Models and Multi-Objective Optimization
General:
The manuscript provides a comprehensive review of optimization methodologies in injection molding, with a particular focus on conformal cooling channels (CCC), surrogate models, and multi-objective optimization techniques. The authors have done an excellent job of synthesizing a vast amount of literature and presenting it in a structured manner. The review is well-organized, and the inclusion of tables summarizing optimization methods and case studies adds significant value.
Comments:
Q1: The review summarized current developments and research focuses on cooling design using numerical methods. However, the reviewer suggested the author to state the gaps or disagreements between the literatures and emphasize the goal of this review aim to address.
ANSWER
The reviewer is correct; the research gap we intend to cover was not clearly stated. We verified that, in some cases, the most recent works in the literature do not consider the best methods used in previous works that optimize operating conditions, feed systems, or cooling systems, for example. Therefore, to strengthen our idea and to complement what we wrote, we added the following sentence to improve our manuscript:
“An evident trend in the body of research on injection molding optimization is the compartmentalized treatment of distinct objectives, such as the design of conformal cooling channels (CCC), as independent challenges. Despite the availability of advanced methodologies, including multi-objective evolutionary algorithms (MOEAs) and surrogate modeling techniques like artificial neural networks (ANNs), which have demonstrated considerable success in optimizing processing parameters, these approaches are seldom leveraged when addressing new or parallel objectives. Instead, many studies regress to elementary strategies, frequently relying on single-objective formulations and classical design of experiments (DOE), thereby reinitiating the optimization framework without building upon prior methodological progress. This review aims to highlight this fragmentation and advocates for a more coherent and cumulative application of advanced optimization paradigms across the various facets of the injection molding process.
Q2: This review could benefit more if the author could expand the discussions about the trade-offs between different methods or algorithms (e.g.: Kringing vs. ANN for surrogate).
ANSWER
The performance of each technique depends on its application, i.e., its data structure. In section 2.6, the analysis required by the reviewer was already made in the original manuscript (page 13). See, for example, the paragraph that finishes with “The effectiveness of each technique depends on the problem’s structure, data availability, and computational resources.”.
Q3: The authors highlighted the importance of sustainability, but this topic is not explored in depth. Could the authors expand on how optimization techniques can contribute to more sustainable manufacturing practices?
ANSWER
A sentence was added into the introduction to stress the importance of sustainability: “Sustainability in injection molding optimization can be enhanced through several strategies, such as reducing cycle time to lower energy usage per part, minimizing waste by decreasing the occurrence of defects, mainly through improved cooling efficiency, optimizing part design to use less raw material without compromising functionality, and adopting process parameters that reduce overall energy consumption. The use of multi-objective optimization algorithms can enable the simultaneous improvement of these objectives.”
Q4: The review discusses various optimization techniques, but are there any case studies or real-world examples where these techniques have been successfully implemented in industry? If so, could the authors include a brief discussion of these cases?
ANSWER
These algorithms are applied in the papers cited in the text. Since this is a review paper, the goal is not to demonstrate the effectiveness of the algorithms discussed, but rather to identify where and for what purpose they are applicable. In this way, we expected to help the reader select the best option for the problem they intend to solve. Most of the studies cited in this review tried to demonstrate the effectiveness of the algorithms referred to here in real-world examples. However, it is challenging to determine whether the results were successfully implemented in industry. This can be the subject of another work.
Q5: The authors briefly mention the potential of machine learning and AI in future research. Could they elaborate on how these technologies could be integrated with traditional optimization methods to create more adaptive and intelligent systems?
ANSWER
Machine learning and AI are based on techniques, such as artificial neural networks (ANN), that are already used and integrated into optimization methodologies for the IM process. These are described in the text, for example, on pages 2-3, 14, 24, and 25. In this case, the combination of topology optimization with machine learning (ML) is identified. Additionally, in these pages, it is demonstrated how these techniques can be integrated with optimization methods.
Q6: For Figure 1B, could the authors add data on the pie chart to indicate the ratio of each injection molding stages?
ANSWER
This change was made.